# Enhanced superconductivity close to a non-magnetic quantum critical point in electron-doped strontium titanate

Yasuhide Tomioka[1], Naoki Shirakawa [1], Keisuke Shibuya [1] & Isao H. Inoue [1]

Studies on quantum critical points (QCP) have focused on magnetic QCPs to date. Remarkable phenomena such as superconductivity due to avoided criticality have been discovered, but we focus here on the non-magnetic counterpart, i.e., the superconductivity of $SrTiO_3$ regarded as being close to a ferroelectric QCP. Here we prepare high-quality $Sr_{1-x}La_xTi(^{16}O_{1-z}{}^{18}O_z)_3$ single crystals without localisation at low temperatures, which allow us to systematically investigate the La substitution of Sr as an alternative to introducing oxygen vacancies. Analysis of our data based on a theoretical model predicts an appearance of the ferroelectric QCP around $3 \times 10^{18}\,cm^{-3}$. Because of the QCP, the superconducting dome of $Sr_{1-x}La_xTiO_3$ can be raised upwards. Furthermore, remarkable enhancement of $T_c$ (~0.6 K) is achieved by $^{18}O$ exchange on the $Sr_{1-x}La_xTiO_3$ crystals. These findings provide a new knob for observing intriguing physics around the ferroelectric QCP.

[1] National Institute of Advanced Industrial Science and Technology (AIST), Tsukuba 305-8565, Japan. Correspondence and requests for materials should be addressed to Y.T. (email: y-tomioka@aist.go.jp) or to I.H.I. (email: isaocaius@gmail.com)

SrTiO$_3$ is one of the most studied transition-metal oxides in the history of condensed matter physics. It is a simple band insulator with a band gap of ~3.3 eV between the Ti 3$d$ and O 2$p$ bands but exhibits various unique and interesting properties and thus remains at the centre of ardent research[1,2]. SrTiO$_3$ undergoes an antiferrodistortive phase transition at ~105 K due to the staggered rotations of TiO$_6$ octahedra around the [001] axis. Many studies suggest that this antiferrodistortive transition suppresses the ferroelectric (FE) phase transition[3,4] at relatively high temperatures, and this scenario is supported by first-principles calculations[5]. However, it remains unclear why the ferroelectricity is suppressed down to very low temperatures (at least 350 mK[6]) despite its phonon structure with polar soft modes remaining. Because of this missing ferroelectricity, a huge static dielectric constant $\varepsilon \sim 24,000$ is observed at low temperatures, resulting in a very large effective Bohr radius of ~0.5 μm. Thus, slight carrier doping of even $2 \times 10^{16}$ cm$^{-3}$ leads to the appearance of an extraordinary dilute metallic state[7]. It is generally believed that the missing ferroelectricity, even at low temperatures, is entirely due to quantum fluctuations: i.e., zero-point motion preventing the complete softening of the transverse optic phonons[6,8]. The low-temperature phase is positioned close to a quantum critical point (QCP), where different phases compete (such as paraelectric, antiferrodistortive, and FE states with similar energies[9–11]). Near the QCP, any residual interactions may drive the system to a superconducting state[8,12,13].

Electron-doped SrTiO$_3$ is one of the most dilute superconductors[14] and has been known for more than half a century; however, its mechanism is poorly understood[15,16]. Several theoretical ideas linking the superconductivity of SrTiO$_3$ to low-temperature instabilities have been proposed. One typical idea is that a soft phonon mode associated with the antiferrodistortive rotation may play a crucial role in the formation of Cooper pairs[17]. Another representative idea involves quantum fluctuations of the FE ordering[6,18,19]. Systematic experimental studies are needed to clarify the mechanism.

Here, we start by demonstrating an appearance of the superconducting dome; i.e., the dome-shaped evolution of $T_c$ as a function of $n$, achieved by La substitution of Sr in SrTiO$_3$ single crystals. Then, we demonstrate the further enhancement of $T_c$ by oxygen isotope ($^{18}$O) exchange of the La-substituted SrTiO$_3$ single crystals. It should be noted here that La-substituted SrTiO$_3$ single crystals have never been studied systematically nor in detail to date. In this work, the optimal $T_c$ reaches 0.44 K at a carrier density $n$ of ~$5.9 \times 10^{19}$ cm$^{-3}$. Moreover, for the $^{18}$O-exchanged La-substituted SrTiO$_3$ single crystals, the maximum $T_c$ is enhanced to 0.55 K at almost the same $n$ of ~$6.0 \times 10^{19}$ cm$^{-3}$. These values of $n$ are slightly lower than $n \sim 1 \times 10^{20}$ cm$^{-3}$, at which SrTiO$_{3-\delta}$ exhibits the optimal $T_c$ ($\delta$ is the amount of oxygen off-stoichiometry that provides two electrons per $\delta$ in a formula unit.)[20,21]. These enhancements of $T_c$ are investigated in this work, and the results suggest a hidden QCP even in a metallic region which may contribute to the $T_c$ enhancement.

## Results

### Resistivity and superconductivity for La-doped SrTiO$_3$.
The temperature dependences of the resistivity $\rho$ for the single crystals of Sr$_{1-x}$La$_x$TiO$_3$ ($x \sim 0.0003$, 0.0005, 0.001, 0.003, 0.005, and 0.007) are presented in Fig. 1a. Each value of $x$ is a nominal value, which was used to prepare each sample (see Methods). As shown in the inset of Fig. 1a, each nominal value of $x$ is almost identical to the number of carriers per Ti site, which was deduced from Hall effect measurements (see Supplementary Figure 7). It is widely accepted that in the oxygen deficient SrTiO$_{3-\delta}$, the carrier doping $\delta$ is directly related to the defect formation on the Ti–O

bond of SrTiO$_3$, whereas in Sr$_{1-x}$La$_x$TiO$_3$, the A-site substitution of the ABO$_3$ perovskite-type structure does not directly introduce disorder to the Ti–O conduction paths[22]. At least for the relatively larger carrier-doping region, La substitution would be an ideal method to investigate the carrier-doping-dependent phenomena of SrTiO$_3$. On the other hand, the disturbance due to the oxygen defects appears to be less significant in the extremely dilute doping region[7,14,23], where the La substitution is not easily controlled. It should be noted, in passing, that the ideal thin film of Sr$_{1-x}$La$_x$TiO$_3$ fabricated by the molecular beam epitaxy method, the mobility reaches 30,000 cm$^2$ V$^{-1}$ s$^{-1}$[24]. But both this large mobility as well as that of our Sr$_{1-x}$La$_x$TiO$_3$ single crystals fit to the general trend of mobility vs. carrier concentration seen in $n$-doped SrTiO$_3$[25] (see Supplementary Note 3).

As apparent from Fig. 1a, the resistivity decreases with increasing La substitution. At 5 K, the resistivity is ~$3 \times 10^{-4}$ $\Omega$cm for $x \sim 0.0005$, which changes to ~$1.6 \times 10^{-4}$ $\Omega$cm for $x \sim 0.007$. These behaviours appear to reflect that of a conventional metal. The resistivity is plotted as a function of $T^2$ in Fig. 1b and c. Although the $T^2$ dependence of the resistivity is in general a representation of a three-dimensional Fermi liquid[26], it should be noted that the $\rho \sim AT^2$ behaviour of the conventional Fermi liquid requires Umklapp scattering, for which the smallest reciprocal lattice vector must not exceed four times the Fermi wave vector[23]. The smallest carrier density that satisfies this condition is greater than $2 \times 10^{20}$ cm$^{-3}$. Therefore, the carrier density of our Sr$_{1-x}$La$_x$TiO$_3$ is too small for Umklapp scattering, indicating that the $\rho \sim AT^2$ behaviour up to high temperatures may be related to other mechanisms[15]. However, it is very intriguing that the behaviour of the coefficient $A$ appears to reflect the Lifshitz transition, which is the change of the number of Fermi surfaces that occurs by changing the carrier density. This means the value of $A$ is related to the fermiology of SrTiO$_3$. It should be noted that this does not underpin the observed $\rho \sim AT^2$ due to the Umklapp scattering in the conventional Fermi liquid. We will discuss this issue later.

Figure 1d shows the temperature dependences of the resistivity below 1 K for the same samples in Fig. 1a measured using a $^3$He/$^4$He dilution refrigerator. Notably, no upturn in resistivity was observed upon decreasing the temperature for any of the samples with different $x$ values, which differs from the reported weak localisation[27] or charge Kondo effect observed in Sr$_{1-x}$Ca$_x$TiO$_{3-\delta}$[28]. The resistive superconducting transitions are clearly observed. In this study, we define the superconducting transition temperature $T_c$ which gives the mid-point of the resistivity during the resistance drop for the superconductivity as described in the Supplementary Note 4. The values of $T_c$ increase from 0.26 K for $x \sim 0.0005$ to 0.41 K for $x \sim 0.003$. As $x$ increases further, $T_c$ decreases to 0.34 K for $x \sim 0.007$. All the results are summarised in Table 1.

### Comparison between $^{18}$O-exchanged and $^{18}$O-free Sr$_{1-x}$La$_x$TiO$_3$.
As described in the Methods section, we prepared two Sr$_{1-x}$La$_x$TiO$_3$ single-crystal rods with the same value of $x$, one of which was $^{18}$O exchanged. For each of the $^{18}$O-exchanged single crystals, the amount of $^{18}$O (the value of $z$) was evaluated from the Raman scattering (see Supplementary Figure 1). The temperature-dependences of the resistivity for Sr$_{1-x}$La$_x$Ti($^{16}$O$_{1-z}$$^{18}$O$_z$)$_3$ with $(x, z) = (\sim 0.002, 0)$, $(\sim 0.002, 0.57)$, $(\sim 0.0035, 0)$, $(\sim 0.0035, 0.57)$, $(\sim 0.01, 0)$, and $(\sim 0.01, 0.60)$ are plotted in Fig. 2a. The $z = 0.57$ and $z = 0.60$ samples are denoted hereafter as $z = 0.6$ for convenience. Apparent $T^2$ dependence up to high temperatures was also observed in these samples, as shown in Fig. 2b.

Figure 2c shows the behaviour of the coefficient $A$ of $\rho \sim AT^2$ for all the Sr$_{1-x}$La$_x$Ti($^{16}$O$_{1-z}$$^{18}$O$_z$)$_3$ samples in Fig. 1a and Fig. 2a as a function of the carrier density $n$. We deduced the value of $A$

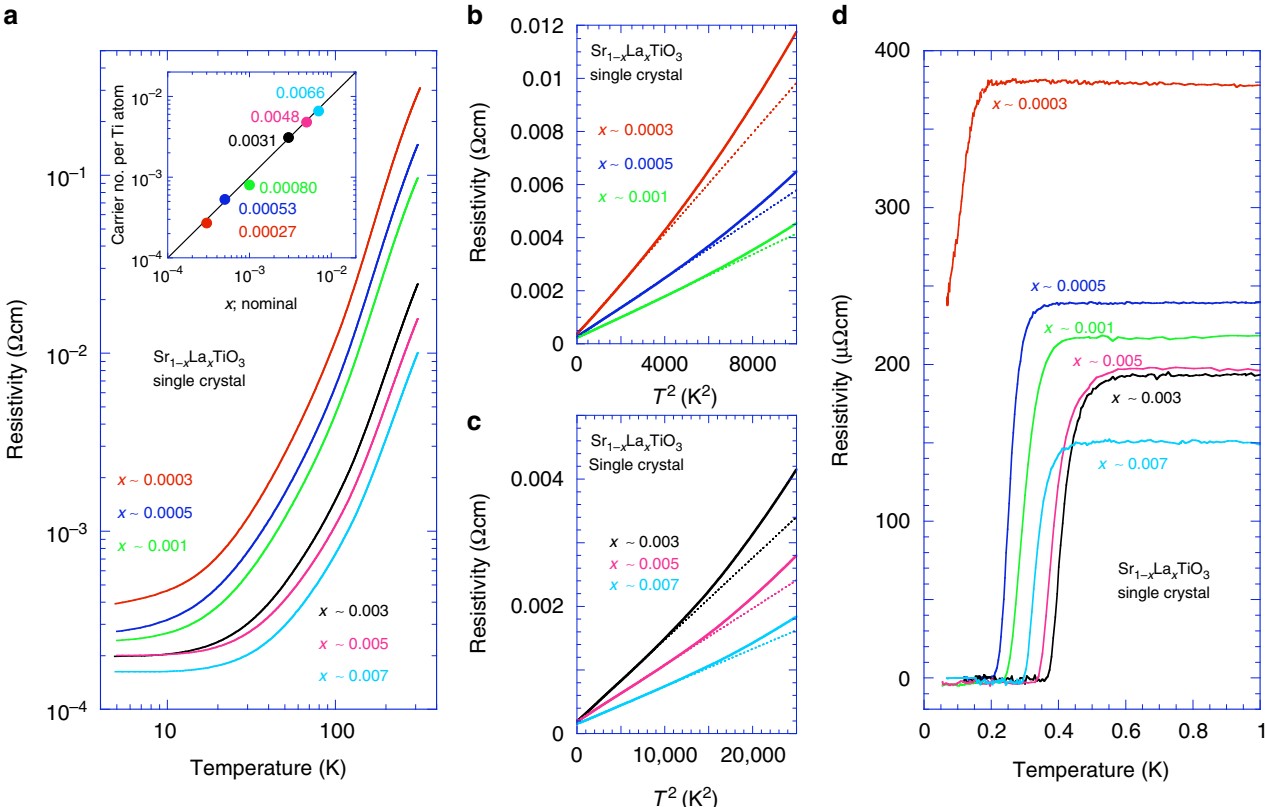

**Fig. 1** Resistivity and superconductivity of $Sr_{1-x}La_xTiO_3$ single crystals. **a** Temperature dependence of resistivity in $Sr_{1-x}La_xTiO_3$ single crystals with nominal values of $x \sim 0.0003$, 0.0005, 0.001, 0.003, 0.005, and 0.007 with which we substituted La for Sr in the raw material. The inset shows that the number of electrons per Ti site determined using the Hall effect measurements was almost identical to the nominal value of $x$. **b, c** Resistivity in **a** re-plotted vs. $T^2$. The straight dotted lines represent the best fits to the data below 40 K. As $x$ increases, the slope gradually decreases, and the deviation point of the straight line from the real data shifts to higher temperatures. **d** Resistivity measured using a $^3He/^4He$ dilution refrigerator plotted against $T$ below 1 K for the same samples shown in **a**. With increasing La substitution, the residual resistivity decreases monotonously, whereas the value of $T_c$ increases up to $x \sim 0.003$ and decreases for $x \sim 0.007$

| **Table 1 Specifications of $Sr_{1-x}La_xTiO_3$ single crystals** | | | | | | | |
|---|---|---|---|---|---|---|---|
| $x$ (nominal) | $n$ (per Ti) | $n$ (cm$^{-3}$) | $\rho$ (300 K) ($\Omega$cm) | $\rho$ (5 K) ($\Omega$cm) | $A$ ($\mu\Omega$cmK$^{-2}$) | $\mu$ (cm$^2$V$^{-1}$s$^{-1}$) | $T_c$ (K) |
| 0.0003 | 0.00027 | $4.54 \times 10^{18}$ | 0.268 | $3.91 \times 10^{-4}$ | 0.948 | 3300 | (<0.05) |
| 0.0005 | 0.00053 | $8.96 \times 10^{18}$ | 0.138 | $2.74 \times 10^{-4}$ | 0.552 | 2540 | 0.26 |
| 0.001 | 0.0008 | $1.34 \times 10^{19}$ | 0.0898 | $2.43 \times 10^{-4}$ | 0.394 | 1910 | 0.29 |
| 0.003 | 0.0031 | $5.21 \times 10^{19}$ | 0.0222 | $1.94 \times 10^{-4}$ | 0.129 | 618 | 0.41 |
| 0.005 | 0.0048 | $8.01 \times 10^{19}$ | 0.0145 | $2.00 \times 10^{-4}$ | 0.089 | 389 | 0.39 |
| 0.007 | 0.0066 | $1.11 \times 10^{20}$ | 0.0093 | $1.62 \times 10^{-4}$ | 0.059 | 347 | 0.34 |

The nominal value of $x$; the carrier density $n$ (per a Ti site and per cm$^3$) determined from Hall effect measurements; the resistivity $\rho$ at 300 K and 5 K; the coefficient $A$ in $\rho \sim AT^2$ below 40 K; the Hall mobility $\mu = n^{-1}e^{-1}\rho^{-1}$ at 5 K, where e is an elementary charge; and the superconducting critical temperature $T_c$ for our $Sr_{1-x}La_xTiO_3$ single crystals

by fitting the resistances of all the samples below 40 K (see Fig. 1b and c) to distinguish the low-temperature $T^2$ behaviour from the high-temperature $T^3$ behaviour[29]. The obtained value of $A$ decreased drastically by approximately two orders of magnitude with increasing carrier density, and a kink was observed near $n \sim 3.9 \times 10^{19}$ cm$^{-3}$. This value of $n$ for the kink is close to the critical carrier density of the Lifshitz transition ($n \sim 4.4 \times 10^{19}$ cm$^{-3}$ by a band calculation[15], and $\sim 3 \times 10^{19}$ cm$^{-3}$ by a quantum oscillation measurement[30,31]), where the Fermi energy enters the third $t_{2g}$ band. Lin et al. reported a similar kink for $A$ in their SrTiO$_{3-\delta}$ samples when the second $t_{2g}$ band starts to be filled[23]. As mentioned above, the coincidence of the kink of $A$ and the Lifshitz transition is not easily acceptable. This is because the $\rho \sim AT^2$ without the Umklapp scattering cannot be explained by the

conventional Fermi liquid theory, and thus $A$ is not necessary to reflect the topology of the Fermi surface. Furthermore, our $Sr_{1-x}La_xTiO_3$ does not show anomaly of $T_c$ at the Lifshitz transition although the density of states as a function of $n$ may have a rapid change at this transition point. These are interesting future problems, which would be a clue for understanding a superconductivity of this system.

Through the $^{18}O$ exchange, the values of $A$ become larger than those of the corresponding $^{18}O$-free ones. If we accept the Fermi liquid-like understanding of $A$, this result indicates that the effective mass of an electron may become larger[32,33], possibly resulting from enhancement of the electron–phonon interactions. Because of the large electron–phonon interactions, the resistivity in the entire temperature range as well as the superconducting $T_c$ are enhanced.

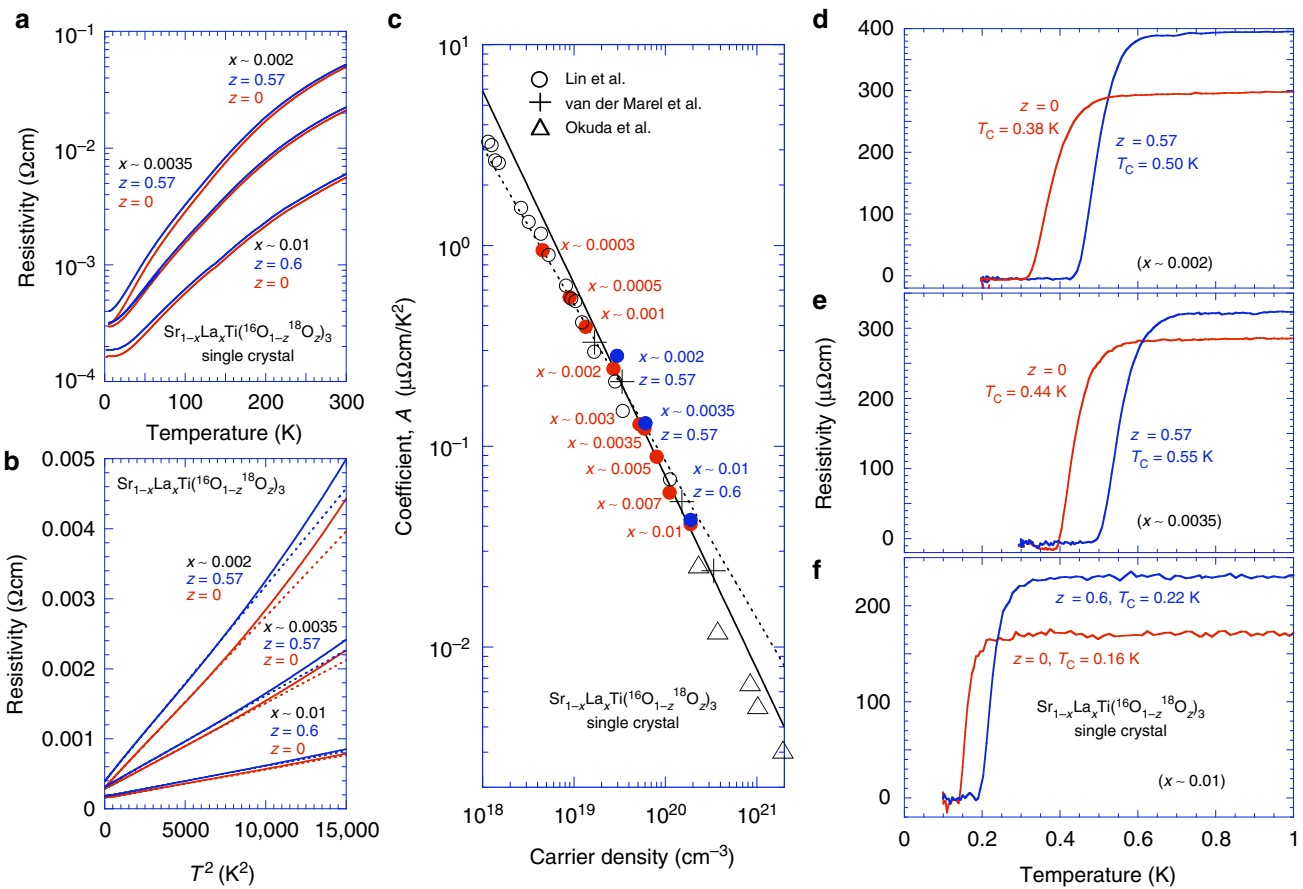

**Fig. 2** Resistivity and superconductivity of $Sr_{1-x}La_xTi(^{16}O_{1-z}^{18}O_z)_3$ single crystals. **a** Temperature dependence of resistivity for $^{18}O$-exchanged $Sr_{1-x}La_xTi$ $(^{16}O_{1-z}^{18}O_z)_3$ with $(x, z)$ = (~0.002, 0.57), (~0.0035, 0.57), and (~0.01, 0.60) (blue lines) compared with that for $Sr_{1-x}La_xTiO_3$ with $x$ ~ 0.002, 0.0035, and 0.01 (red lines). **b** Resistivity $\rho$ vs. $T^2$. The dotted lines represent $\rho \sim AT^2$ relations to fit the experimental data. The coefficient $A$ increases with $^{18}O$ exchange, indicating mass enhancement of the carriers. **c** Logarithmic plot of $A$ as a function of carrier density $n$ for all our $Sr_{1-x}La_xTi(^{16}O_{1-z}^{18}O_z)_3$ samples in Fig. 1a and **a** with the data in the literature: $SrTiO_{3-\delta}$[23] (open circles), $SrTi_{1-x}Nb_xO_3$[15] (crosses), and $Sr_{1-x}La_xTiO_3$[26] (triangles). The straight lines are guides to the eye. A two-orders-of-magnitude drop of $A$ for increasing $n$ is observed, and a kink appears at $n \sim 4 \times 10^{19}$ cm$^{-3}$ near the critical value for the Lifshitz transition[15,30,31]. **d–f** Residual resistivity $\rho_0$ and superconducting transition temperatures $T_c$ measured using a $^3He/^4He$ dilution refrigerator below 1 K for the same samples shown in **a**

Figure 2d–f shows the temperature dependence of the resistivity below 1 K for the same single crystals shown in Fig. 2a. The superconducting transition temperatures of the $^{18}O$-exchanged and $^{18}O$-free samples were compared for the $x$ ~ 0.002, 0.0035, and 0.01 samples. All the experimentally deduced parameters for Fig. 2 are summarised in Table 2.

**Superconducting dome for $Sr_{1-x}La_xTi(^{16}O_{1-z}^{18}O_z)_3$.** In Fig. 3a, $T_c$ is plotted as a function of $n$ for our $Sr_{1-x}La_xTi(^{16}O_{1-z}^{18}O_z)_3$ single crystals prepared using the floating zone (FZ) method. For the $z = 0.4$ samples, see the Supplementary Note 4. In addition to our data, reported values of $T_c$ for $SrTiO_{3-\delta}$[21,28,31] and $SrTi_{1-x}Nb_xO_3$ single crystals[31,34] as well as La-substituted $SrTiO_3$ single crystals[27] are also included in the plot; all of these reported samples in the literature were prepared using the Verneuil method. Despite the different crystal growth procedures (FZ or Verneuil), the values of $T_c$ for all the La-substituted $SrTiO_3$ single crystals ($z = 0$) comply with a single superconducting dome. At $x$ ~ 0.0035 ($n$ ~ $6 \times 10^{19}$ cm$^{-3}$), the dome reached a maximum $T_c$ of ~0.44 K.

## Discussion
FE fluctuations in $SrTiO_3$ with zone-centre soft-mode optical phonons (either longitudinal or transverse) have been considered to play some relevant roles in the mechanism of the superconductivity[16,35–39]. As the FE fluctuations are clearly suppressed with increasing carrier density $n$ (as the system becomes more metallic), the concomitant disappearance of the superconductivity for the overdoped region ($n \gtrsim 2 \times 10^{20}$ cm$^{-3}$) may be an implication of the superconductivity driven by the ferroelectricity. On the other hand, in the underdoped region ($n \lesssim 1 \times 10^{20}$ cm$^{-3}$), the FE fluctuations are enhanced by the decrease of $n$. However, if $n$ is too small, the superconductivity is depressed, as the carrier density is not sufficient to provide robust superconductivity. The formation of the superconducting dome can be explained in this way, though there is room for argument.

What is prominent in Fig. 3a is not only the elucidation of the superconducting dome of $Sr_{1-x}La_xTiO_3$ but also the large enhancement of its $T_c$ by the $^{18}O$ exchange. As described above, there are many models of the mechanism of superconductivity in $SrTiO_{3-\delta}$; however, most of these models cannot be simply applied to our experimental data to provide insight into the enhancement by the $^{18}O$ exchange. A recent theoretical approach proposed by Edge et al.[18] is one of the most tractable approaches, such that we attempted to compare this model to our data as follows. It should be noted here that we do not rule out any other theories which may explain the rise of the superconducting dome.

We calculated the theoretical $T_c$ vs. $n$ curve applying the model in ref. [18] to the reported[21] and our experimental $T_c$ vs. $n$ data. In

**Table 2 Specifications of $^{18}$O-exchanged and $^{18}$O-free single crystals**

| x (nominal) | n (per Ti) | n (cm$^{-3}$) | $\rho$ (300 K) ($\Omega$cm) | $\rho$ (5 K) ($\Omega$cm) | A ($\mu\Omega$cmK$^{-2}$) | z | $T_c$ (K) |
|---|---|---|---|---|---|---|---|
| 0.002 | 0.0016 | 2.70 × 10$^{19}$ | 0.0495 | 3.08 × 10$^{-4}$ | 0.249 | 0 | 0.38 |
| | 0.0018 | 2.96 × 10$^{19}$ | 0.0529 | 3.99 × 10$^{-4}$ | 0.281 | 0.57 | 0.50 |
| 0.0035 | 0.0036 | 5.88 × 10$^{19}$ | 0.0210 | 2.95 × 10$^{-4}$ | 0.124 | 0 | 0.44 |
| | 0.0036 | 6.04 × 10$^{19}$ | 0.0224 | 3.19 × 10$^{-4}$ | 0.132 | 0.57 | 0.55 |
| 0.01 | 0.011 | 1.88 × 10$^{19}$ | 0.00599 | 1.75 × 10$^{-4}$ | 0.041 | 0 | 0.16 |
| | 0.011 | 1.88 × 10$^{19}$ | 0.00693 | 2.34 × 10$^{-4}$ | 0.044 | 0.60 | 0.22 |

The nominal value of x, the carrier density n (per a Ti site and per cm$^3$) determined from Hall effect measurements, the resistivity $\rho$ at 300 K and 5 K, the coefficient A in $\rho$ - $AT^2$ below 40 K, the amount of $^{18}$O exchange z estimated from Raman scattering spectroscopy, and the superconducting critical temperatures $T_c$ for our Sr$_{1-x}$La$_x$Ti($^{16}$O$_{1-z}$$^{18}$O$_z$)$_3$ single crystals

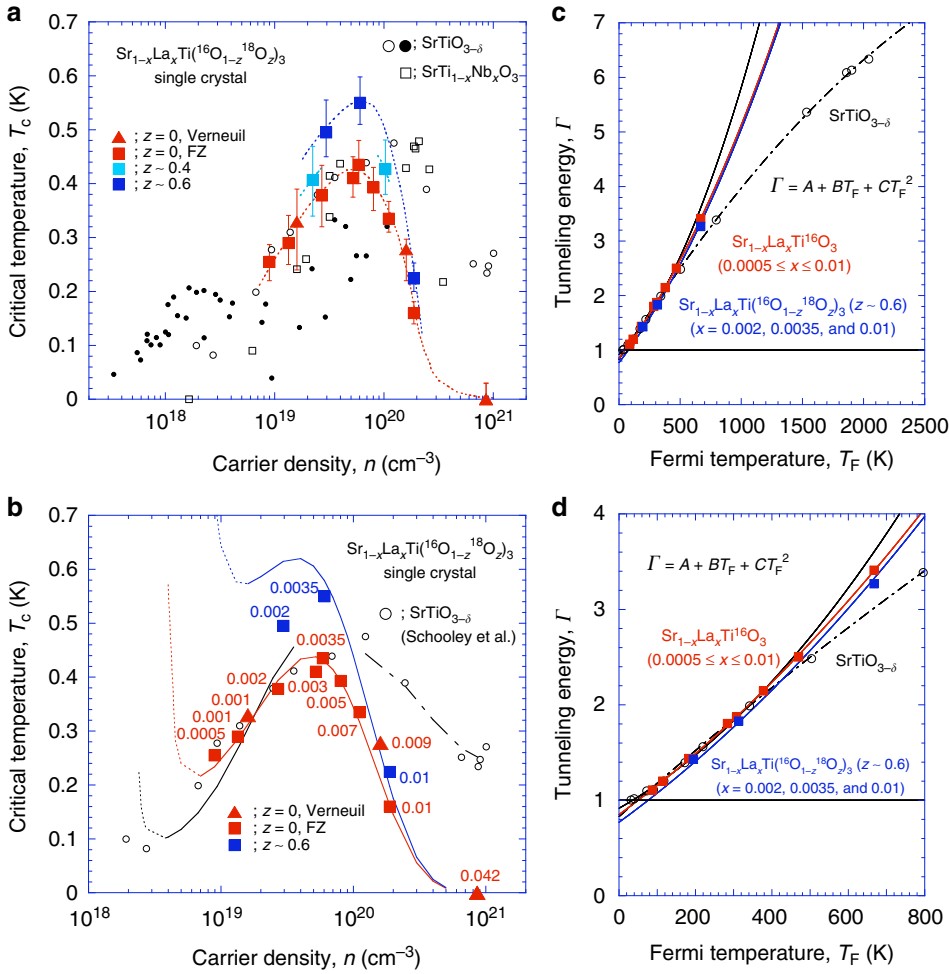

**Fig. 3** Evolution of the superconducting dome with La substitution and $^{18}$O exchange. **a** $T_c$ vs. $n$ plot for the Sr$_{1-x}$La$_x$Ti($^{16}$O$_{1-z}$$^{18}$O$_z$)$_3$ single crystals shown in Figs. 1a and 2a. The upper and lower error bars are determined from the onset and end temperatures of the superconductivity (see Supplementary Figure 4). $T_c$ for SrTiO$_{3-\delta}$[21,28,31], SrTi$_{1-x}$Nb$_x$O$_3$[31,34], and La-substituted SrTiO$_3$[27] are also plotted. **b** The lines were obtained from calculations based on the model in ref. [18], which are compared with some of the experimental data in **a**. **c** Tunnelling energy $\Gamma$ vs. Fermi temperature $T_F$ deduced from the experimental values of $T_c$ and $n$ using Supplementary Equations 1 and 2: Sr$_{1-x}$La$_x$TiO$_3$ (red squares) and Sr$_{1-x}$La$_x$Ti($^{16}$O$_{1-z}$$^{18}$O$_z$)$_3$ ($z = 0.6$; $x \sim 0.002$, 0.0035, and 0.01) (blue squares) from this study; SrTiO$_{3-\delta}$ (black circles) from the literature[21]. The experimental data were fitted by the lines of $\Gamma = A + BT_F + CT_F^2$. From these fitting results for $\Gamma$, we obtained the $T_c$ vs. $n$ relation represented by the lines in **b**. **d** The blow-up of **c** to emphasise where the $\Gamma$ crosses $\Gamma = 1$ line

Fig. 3b, the curves (solid and partially dashed lines) are plotted with the experimental data points. The black line is for SrTiO$_{3-\delta}$, the red line is for Sr$_{1-x}$La$_x$TiO$_3$, and the blue line is for Sr$_{1-x}$La$_x$Ti($^{16}$O$_{0.4}$$^{18}$O$_{0.6}$)$_3$. Details of the calculation steps are provided in Supplementary Note 7. The model explains fairly well the observed superconducting dome of Sr$_{1-x}$La$_x$TiO$_3$.

The $T_c$ vs. $n$ lines in Fig. 3b are based on the $\Gamma$ vs. $T_F$ plots in Fig. 3c. Here, $\Gamma$ is the tunnelling energy of the double-well potential[18] in analogy with that of magnetic phase transitions, and $T_F$ is the Fermi energy but we used the value of $T_F$ deduced from the carrier density by assuming a free electron gas without mass enhancement. The $\Gamma$ vs. $T_F$ plot was obtained directly from

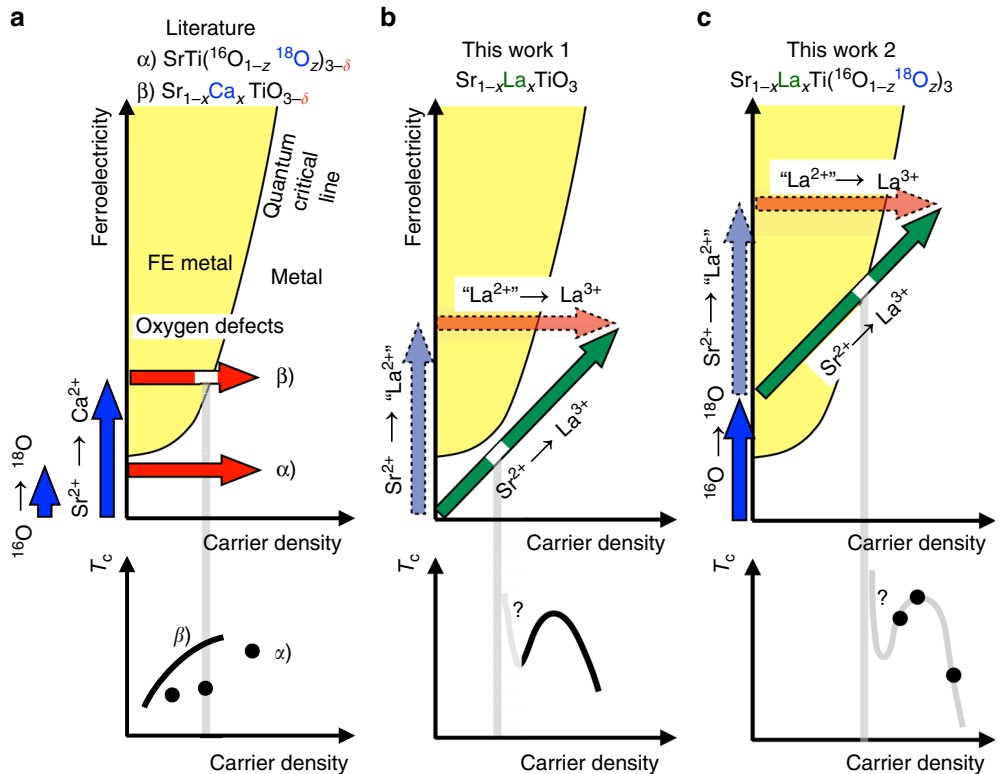

**Fig. 4** Schematic diagrams of La substitution and $^{18}$O exchange. Schematic explanation of the evolution of QCP in the metallic state, carrier doping by oxygen defects (red arrows), $^{18}$O exchange and Ca substitution toward ferroelectricity (blue arrows), as well as the La substitution (green arrow) on the "ferroelectricity" vs. "carrier density" plane. The corresponding $T_c$ vs. $n$ schematics are also shown. **a** $\alpha$) SrTi($^{16}O_{1-z}{}^{18}O_z$)$_{3-\delta}$[44] and $\beta$) Sr$_{1-x}$Ca$_x$TiO$_{3-\delta}$[28], **b** Sr$_{1-x}$La$_x$TiO$_3$, and **c** Sr$_{1-x}$La$_x$Ti($^{16}O_{1-z}{}^{18}O_z$)$_3$. QCP corresponds to the white portion of the arrow at which the behaviour of $T_c$ becomes unpredictable[18]. However, the shift of QCP towards larger carrier-density region is expected to raise the superconducting dome upwards[18,31,42]

the experimental $T_c$ vs. $n$ data, as described in the Supplementary Note 7. Then, we fit the data using a power expansion of $\Gamma$ in $T_F$ such as $\Gamma = A + BT_F + CT_F^2$, where $A$, $B$, and $C$ are fitting parameters. The QCP, i.e., the divergence of $\lambda$, corresponds to the point at which $\Gamma = 1$. It should be noted that our systematic investigation of La substitution made a reliable fitting possible. Then, it has finally elucidated that $\Gamma < 1$ below $n < n_c$ in Sr$_{1-x}$La$_x$TiO$_3$ as seen in Fig. 3b (the blow-up is plotted in Fig. 3d); this means the model predicts the appearance of a QCP at $n = n_c \sim 3 \times 10^{18}$ cm$^{-3}$.

At $n = n_c$, the superconductivity coupling constant $\lambda$ diverges, and $T_c$ at the QCP becomes unpredictable[18]. Our own numerical calculations following the model produced an exponential enhancement of $T_c$ at $n = n_c$ (dotted lines in Fig. 3b). It has been suggested that if QCP is above around $1 \times 10^{18}$ cm$^{-3}$, $T_c$ goes up when the system approaches the QCP[40]. However, the validity of this divergence is controversial. The value of $T_c$ might be suppressed to zero because the model is not simply applicable in the vicinity of QCP. Unfortunately, for our Sr$_{1-x}$La$_x$TiO$_3$ system, it is difficult to explore such an extremely dilute region, where the system actually crosses the QCP. This is an important problem to be clarified by more detailed experimental investigation, such as the application of tensile stress, which is considered to shift the QCP towards the higher doping region leading to a drastic enhancement of $T_c$ at the QCP[41,42] or the isostatic pressure to diminish the QCP[40,43] and suppress $T_c$.

We further attempted to fit the $\Gamma$ vs. $T_F$ data of Sr$_{1-x}$La$_x$Ti ($^{16}O_{1-z}{}^{18}O_z$)$_3$ ($z = 0.6$) using $\Gamma = A + BT_F + CT_F^2$ with the same $B$ and $C$ values as those for Sr$_{1-x}$La$_x$TiO$_3$ (blue dashed line in Fig. 3b). Although this assumption is naive, we do not think that it is unreasonable, because in principle $^{18}$O exchange does not

affect $T_F$. Because our experimental data are only three points, the fitting was not very good. We understand that $n_c$ increased; i.e., the QCP shifts towards larger carrier density with the $^{18}$O exchange. The observed further enhancement of $T_c$ (blue line in Fig. 3a) can be explained in this way by the shift of QCP.

It should be mentioned that the SrTiO$_{3-\delta}$ data[21] in Fig. 3b cannot be simply fitted by the same model in ref. [18]. It seems that the data might be separated into at least two parts: a smaller carrier-density region below ~$5 \times 10^{19}$ cm$^{-3}$, and a larger carrier-density region above it. The fitting of the experimental data in the larger carrier-density region (dash-dotted line in Fig. 3b) suggests there is QCP even for SrTiO$_{3-\delta}$ at almost the same carrier density as that of Sr$_{1-x}$La$_x$TiO$_3$ (see the $\Gamma = 1$ point of the red solid line and the dash-dotted line in Fig. 3c and d). On the other hand, in the small carrier-density region, although the data of SrTiO$_{3-\delta}$ are almost identical to those of Sr$_{1-x}$La$_x$TiO$_3$, the two additional points in the lowest carrier density make the fitting worse (see the black solid and dotted line). The analysis predicts the QCP may be close to $2 \times 10^{18}$ cm$^{-3}$.

Figure 4 presents schematic diagrams showing the evolution of the FE QCP in the metallic region. The planes are spanned by the horizontal axes of carrier density and the vertical axes of ferroelectricity. The FE state must be rapidly suppressed by the screening due to mobile carriers; however, a certain ordered state was considered to remain even in the metallic phase for very small $n$[18,28]. Hence, the boundary of the ordered FE state is assumed to penetrate into the carrier doped region as shown in Fig. 4[18].

In the literature, there were two types of investigations of the superconductivity on this plane (Fig. 4a). One is to do $^{18}$O exchange (blue vertical arrow) and to dope carriers by oxygen-

defect creation (red horizontal arrow)[44]. The other is to do $Ca^{2+}$ substitution for $Sr^{2+}$ (blue vertical arrow) and to dope carriers by oxygen-defect creation (red horizontal arrow)[28]. The former did not actually cross the QCP but an enhancement of $T_c$ was observed. The latter crossed the QCP but there was no anomaly in the $T_c$ vs. $n$ behaviour at QCP, contradicting to the theoretical models[18,42]. However, in these systems, the values of $T_c$ are relatively large even in the small carrier-density region. There must be a non-negligible contribution of the latent ferroelectricity in the metallic state to the enhancement of the superconductivity.

The La substitution process can be expressed as the diagonal line (Fig. 4b). As discussed in the Supplementary Note 6, if we consider a virtual "$La^{2+}$" substitution as an analogy of $Sr_{1-x}Ba_xTiO_3$ and $Sr_{1-x}Ca_xTiO_3$, it is reasonable to assume that the La substitution process can be decomposed into 1) $SrTiO_3$ to $Sr_{1-x}$"$La^{2+}$"$_xTiO_3$ process (light blue vertical arrow) and 2) $Sr_{1-x}$"$La^{2+}$"$_xTiO_3$ to $Sr_{1-x}La_xTiO_3$ process (light red horizontal arrow). Therefore, the actual La substitution process is represented by the diagonal green arrow. (This green arrow is not necessary to be the straight one but for simplicity we consider it is a simple straight arrow.) Since there is no experimental evidence, we are not sure the diagonal line crosses the border of the FE metal and the normal metal (i.e., the so-called quantum critical line QCL), but the analysis of our experimental data with the theoretical model[18] has predicted the appearance of QCP. Although the figure is schematic, we can assume the crossing point (or the point where the diagonal line is in the vicinity of the border line) is located in the low carrier-density region.

The $SrTi(^{16}O_{1-z}{}^{18}O_z)_3$ exhibits ferroelectricity at $z = 0.36$ (refs. [45]. and [46]). Then, the La substitution process for $Sr_{1-x}La_xTi(^{16}O_{1-z}{}^{18}O_z)_3$ ($z > 0.36$) corresponds to the green diagonal line in Fig. 4c. This diagonal line is sure to cross the border of the FE metal and the normal metal. The important argument in this schematic diagram is that the crossing point for $Sr_{1-x}La_xTi(^{16}O_{1-z}{}^{18}O_z)_3$ ($z > 0.36$) should appear at larger $n$ than that of $Sr_{1-x}La_xTiO_3$, explaining why $T_c$ of $Sr_{1-x}La_xTi(^{16}O_{1-z}{}^{18}O_z)_3$ ($z > 0.36$) is higher than that of $Sr_{1-x}La_xTiO_3$. It is clearly demonstrated that diagonal La substitution $x$ in $Sr_{1-x}La_xTi(^{16}O_{1-z}{}^{18}O_z)_3$ is much more effective for increasing $T_c$ than the simple horizontal oxygen-defect doping $\delta$ in $SrTi(^{16}O_{1-z}{}^{18}O_z)_{3-\delta}$.

In summary, we prepared high-quality $Sr_{1-x}La_xTi(^{16}O_{1-z}{}^{18}O_z)_3$ single crystals in the low doping region ($0.0003 \lesssim x \lesssim 0.01$) with oxygen-isotope exchange. La substitution is an ideal method to introduce carriers into $SrTiO_3$; our samples do not show tendencies of localisation at low temperatures. The carrier density is too small for Umklapp scattering[23]; however, the $\rho \sim AT^2$ behaviour is seen and the coefficient $A$ reflects the topological change of the Fermi surface, suggesting that the $Sr_{1-x}La_xTiO_3$ system cannot be simply understood by a conventional Fermi liquid theory. The superconducting $T_c$ exhibits a dome shape similar to that of $SrTiO_{3-\delta}$. The values of $T_c$ were remarkably enhanced to 0.55 K for $Sr_{1-x}La_xTi(^{16}O_{1-z}{}^{18}O_z)_3$ ($x \sim 0.0035$ and $z \sim 0.6$). Although the mechanism of the superconductivity in $SrTiO_{3-\delta}$ and related materials remains a controversial topic, we demonstrated reasonable agreement of our experimental data with the hidden FE QCP model; i.e., the QCP is located in a region where the system is already metallic and will contribute to the $T_c$ enhancement.

Ferroelectricity is a state of broken inversion symmetry. If we can prepare a sample within the QCP of $Sr_{1-x}La_xTi(^{16}O_{1-z}{}^{18}O_z)_3$, it may be a novel realisation of noncentrosymmetric superconductivity[47], which is currently under intensive study, as it may hold mixed-parity pairing mechanisms with topological aspects to their superconducting states. Such a sample will possibly incubate extremely large and highly anisotropic upper critical fields and topologically protected spin currents[48,49]. Our findings will add another page to current research on the FE QCP and associated superconductivity.

## Methods

**Synthesis of $Sr_{1-x}La_xTiO_3$ single crystals**. Mixed powders of $SrCO_3$, $La_2O_3$, and $TiO_2$ in a ratio of $1-x{:}x/2{:}1$ were calcined at 500 °C in air for 2–3 h. The calcined powders were sintered at 1000 °C in air for 5 h. Then, the powders were pulverised and formed into a rod with a diameter of approximately 4 mm and length of approximately 60 mm. The rod was fired at 1300 °C–1380 °C for 5 h in a flowing argon gas atmosphere. The crystal growth of $Sr_{1-x}La_xTiO_3$ was performed with a conventional FZ method. In this method, we use a furnace equipped with double hemi-ellipsoidal mirrors coated with gold. Two halogen incandescent lamps were used as heat sources. The crystals were grown in a stream of argon gas, and the growth rate was set at 10–15 mm per hour.

**Synthesis of $Sr_{1-x}La_xTi(^{16}O_{1-z}{}^{18}O_z)_3$ single crystals**. Because the oxygen diffusion in La-substituted single crystals is extremely slow[50], the oxygen isotope ($^{18}O$) exchange was fulfilled for the mixed powders of $SrTiO_3$, $La_2O_3$, and $TiO_2$. We first prepared the $SrTiO_3$ powders. The mixed powder of $SrCO_3$ and $TiO_2$ with a molar ratio of 1:1 was calcined at 500 °C for 2–3 h. The calcined powders were sintered at 1280 °C for 30 h in air to synthesise the polycrystalline powders of $SrTiO_3$. Then, we mixed the powders of $SrTiO_3$, $La_2O_3$, and $TiO_2$ in a ratio of $1-x{:}x/2{:}x$ and formed them into a rod with a diameter of 4 mm and length of 60 mm. The rod was sintered at 1000 °C in air for 5 h. The sintered rod was then divided into two rods. In a furnace equipped with double sapphire tubes, one rod was placed in the tube with flowing $^{16}O_2/^{18}O_2$ mixed gas atmosphere to exchange the $^{16}O$ atoms with $^{18}O$ ones. The other rod was placed in the other tube in a flowing oxygen ($^{16}O$) atmosphere as a reference. The temperature of the furnace was then increased to 1000 °C to promote the $^{16}O/^{18}O$ exchange. The annealing at 1000 °C was continued until the equilibrium concentration of the $^{16}O_2/^{18}O_2$ atmosphere was reached. The amount of $^{18}O$ atoms absorbed into the rod was by first estimated from the weight change of the rod and then from the amount of $^{18}O_2$ gas in the atmosphere of the furnace, which was determined using a quadrupole mass analyser. These two values matched very well, which indicated that the $^{18}O$-exchange was sufficiently controlled in our experiments. Both the $^{18}O$-free and $^{18}O$-exchanged feed rods were used for the single crystal growth, the procedure for which was the same as described in the previous section. The amount of $^{18}O$ atoms in the grown crystals was finally determined from the frequency shift of the Raman scattering (Supplementary Figure 1).

**Resistivity measurements**. The formation of a single crystal was confirmed using the reflection Laue method. Several pieces were cut from the single crystal rod along the [100] direction of the cubic indices. The pieces were further formed into a rectangular shape with the longest dimension parallel to the [100] direction of the cubic indices. The typical dimensions were $0.5 \times 0.3 \times 7\ mm^3$. The d.c. resistivity $\rho$ and Hall resistivity $\rho_H$ for $5\ K \leq T \leq 300\ K$ were measured in a cryostat equipped with a superconducting magnet (Physical Property Measurement System (PPMS), Quantum Design Inc.). The resistivity down to 50 mK was measured using an a.c. resistance bridge (LR700, Linear Research Inc.) in a cryostat with a $^3$He/$^4$He dilution refrigerator ($\mu$ dilution, Taiyo-Toyo Sanso Inc.). The electrodes were prepared using ultrasonic indium soldering. The transport current was injected parallel to the [100] direction of the cubic indices.

## Data availability

The data that support the plots within this paper and other findings of this study are available in figshare with the digital object identifier https://doi.org/10.6084/m9.figshare.7434221 (ref. [51]). Further data and resources in support of the findings of this study are available from the corresponding authors upon reasonable request.

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

## Acknowledgements
This study was supported by Japan Society for the Promotion of Science (JSPS) KAKENHI Grant No. 17684020 (Young Scientists A) and Grant Nos. 24244062, 15H02113, and 18H03686 (Category A). The authors would like to thank Yoshihiro Aiura and the late Hiroshi Bando for use of their patented isotope-gas-circulating furnace before the patent was filed and for helping us adapt the equipment for this research project. The authors are also grateful to Jun'ya Tsutsumi and Nao Takeshita for their help with the experiments and to Yasuhiro Tada, Masaki Oshikawa, Justin Ye, Akihito Sawa, Izumi Hase, Matthew Coak, Alexander Balatsky, and Kamran Behnia for fruitful discussions.

## Author contributions
The single crystal growth and resistivity measurements were performed by Y.T. Oxygen isotope substitution was performed by I.H.I. The ³He/⁴He dilution refrigerator was set up by N.S., and all the measurements below 1 K were conducted by Y.T. and N.S. Raman scattering spectra were measured and analysed by K.S. All the authors discussed the results and wrote the manuscript.

## Additional information

**Competing interests:** The authors declare no competing interests.

