## [Peer Review File · Nature Communications]

Reviewers' comments:

Reviewer #1 (Remarks to the Author):

This paper reports on a systematic study of the normal-state transport and superconducting resistive transitions in La-doped strontium titanate and its evolution with ¹⁸O enrichment. The authors not only confirm the existence of an inverse isotopic effect, but also succeed (for the first time) to enhance the superconducting T_c well-above its peak in the absence of ¹⁸O. Since ¹⁸O is known to stabilize ferroelectricity in undoped STO, the results reported in this study are a spectacular confirmation of an intimate link between superconductivity and ferroelectricity in this system, in agreement with the ferroelectric quantum-critical scenario.

I think the paper deserves publication in Nature Communications, provided that the authors correct inaccuracies and clarify ambiguities, which can be found in the present version. Here are my concerns.

1. Distinguishing between bulk and resistive superconducting transitions- According to the abstract: "Compared with the same material doped by oxygen vacancies, the La-substituted SrTiO₃ exhibited a much higher T_c at the same carrier densities." This is an incorrect statement based on a fallacious comparison. Indeed, Fig. 3a compares the critical temperatures in La-doped STO, obtained by RESISTIVITY with critical temperatures of Nb-doped and oxygen-reduced STO reported by MAGNETIZATION in ref. 22. Now, the same group of authors (Koonce, Cohen, Schooley, Hosler & Pfeiffer published another paper reporting on resistive transitions (See Phys. Rev. Lett. 14, 305 (1965)). Comparing them one can see that while for resistive transition T_c peaks to 0.4 K, for bulk magnetization the maximum T_c is 0.3 K). The difference between bulk and resistive T_c s was discussed in detail in recent literature (See for example Fig. 1 ref. 36 and Fig. 3 in arXiv:1804.07067). The authors should amend their paper by comparing their resistive T_c s with the resistive T_c of Nb-doped and oxygen-reduced STO.
2. La substitution and ferroelectricity- The following statement in the abstract appears to be inaccurate or unjustified: "La substitution drives the system towards the ferroelectric QCP." Measurements of dielectric constant and Raman spectroscopy have shown that substituting ¹⁶O with ¹⁸O or Sr with Ca makes the solid ferroelectric. Is there any evidence that this is also the case by substituting Sr with La? Is there any evidence of the hardening of the soft TO mode in the La-substituted samples? Since the authors have performed Raman scattering, they can address this question.
3. Isotopic dependence of the magnitude of the T-square prefactor- The authors report that A become larger with ¹⁸O substitution. However, what one sees in Fig.2b is an enhancement in resistivity well above the temperature window where resistivity plotted vs. T^2 is linear. The figure should be zoomed up to make this clear and the discussion should distinguish between the low-temperature T-square and the high-temperature T-cube (reported and discussed in Phys. Rev. 155, 796 (1967); NPJ Quantum Materials 2: 41 (2017) and arXiv:1806.05775 (2018)) regimes.
4. Mobility vs. carrier concentration in La-substituted strontium titanium- In thin films of La-doped Strontium titanate, carrier mobility can become as large as 30000 cm²/V.s. (See Nat. Mater. 9, 482 (2010)). The authors should comment on the fact that according to their Fig. 1D, the residual resistivity of their most underdoped sample ($x=0.0003$) appears to point to a much lower mobility. Does this mobility fit in to the general trend of mobility vs. carrier concentration seen in n-doped strontium titanate (See Fig. 4 in J. Phys.: Condens. Matter, 27 375501 (2015)).

Reviewer #2 (Remarks to the Author):

In this paper the authors study superconductivity in electron-doped strontium titanate (STO) near a ferroelectric quantum critical point. They do so by doping the A site with lanthanum and by isotope substitution of the oxygen. With this approach the authors are able to achieve higher peak

superconducting transition temperatures than those achieved previously.

I certainly find this study very intriguing, but would like some more information to put it in context of prior work. More specifically I have the following comments/questions for the authors:

1. In the abstract and then in the main text the authors write that lanthanum substitution both dopes STO and drives it towards a ferroelectric quantum critical point. They also note that La-doped STO crystals have not been previously studied systematically to date. Transport and Hall measurements support their statement about doping. It is known that oxygen substitution and application of external pressure can drive STO to its ferroelectric quantum critical point, but I do not see a reference or data to support the authors' assertion here that La-substitution has a similar effect. More generally it would be very useful, particularly for comparison to previous work, to have a phase diagram for La-doped for oxygen-16 before considering the isotopically mixed case.
2. The authors observe AT^2 behavior of the resistivity and ascribe it to Lifshitz transitions, as was done earlier by Lin and coworkers. However it's not clear how this interpretation accounts for the observed temperature-dependence. It would be very helpful if the authors could expand their A vs. carrier density plot to compare their results with those found in other doped STO systems.
3. Naively one expects the superconducting transition temperature to be enhanced across a Lifshitz transition due to the availability of additional states. Mysteriously the opposite (T_c decreases) occurs in the measurements by Lin et al., and it has been suggested that this behavior may be due to disorder. What is the behavior of T_c as a function of n here as bands cross the Fermi level?
4. In Figure 3 the authors have compared their T_c vs n results to several previous studies. They suggest the importance of ferroelectric quantum criticality in maximizing the superconducting transition temperature. Their conclusions are similar to those from recent pressure-driven Nb-doped STO measurements (arXiv 1801.08121) and links with this work should be discussed.

In summary, the authors are presenting a new materials pathway towards exploring the mysterious superconductivity in doped strontium titanate. Their studies, combined with parallel and prior work, will help the community distinguish the extrinsic and the intrinsic aspects of this rich problem.

Reviewer #3 (Remarks to the Author):

This paper reports the enhancement of superconducting transition temperatures in oxygen-isotope-exchanged SrTiO₃ and La-doped SrTiO₃ systems. The authors have found out the change in the relationship between superconducting T_c and enhancement of oxygen mass. The authors have prepared many samples including La and 18O and got several important results in the superconductivity. The findings by the authors are interesting but some of the important explanations are seemed lacking. Due to this reason, the reviewer thinks that the authors should modify the article to be more sophisticated one before the review. Following points are raised as problems in the present version.

(1) First, the authors have not given the precision of the experiments. At least, they should give error bars in all the figures. Significant digits for all the data should be given in the text and figures, because very small change in the resistivity is discussed throughout the article. The reviewer cannot be convinced that all results are worthy for the detailed discussions.

(2) Sample preparation: In supplemental section. The authors have determined the isotope composition of the sample by the Raman spectroscopy. The authors have given no direction of the

sample and optical setup used for the spectrum measurements. Also, reasonableness of the isotope composition determined by the Raman peak shift should be confirmed by comparing the result done by other method. Cross checking should be done to confirm the accuracy of the experiments.

(3) Fig.2(c): The oxygen-isotope-exchanged sample seems lying on a straight line. Does oxygen-isotope-exchanged system show a Lifshitz transition or not?

(4) Fig.2(d): The result of AC susceptibility should be moved to Supplemental sections. It is confusing to show as an insert.

(5) Fig.3(c): What is the definition of ferroelectricity? The authors should explain the detail of the meaning although this figure is too schematic. What is the physical meaning of the straight lines? How did the authors draw the OCL curves? These points should be explained clearly.

Reply to Reviewer #1

We would like to thank the Reviewer #1 for the careful reading of our manuscript. We are happy he/she considers our work “succeed (for the first time) to enhance the superconducting T_c well-above its peak in the absence of 18O”, “the results reported in this study are a spectacular confirmation of an intimate link between superconductivity and ferroelectricity”, and our “paper deserves publication in Nature Communications”. We have taken into account the remarks made by the Reviewer #1 to improve the clarity of our presentation.

Comment 1, Reviewer #1

1. Distinguishing between bulk and resistive superconducting transitions- According to the abstract: “Compared with the same material doped by oxygen vacancies, the La-substituted SrTiO₃ exhibited a much higher T_c at the same carrier densities. This is an incorrect statement based on a fallacious comparison. Indeed, Fig. 3a compares the critical temperatures in La-doped STO, obtained by RESISTIVITY with critical temperatures of Nb- doped and oxygen-reduced STO reported by MAGNETIZATION in ref. 22. Now, the same group of authors (Koonce, Cohen, Schooley, Hosler & Pfeiffer published another paper reporting on resistive transitions (See Phys. Rev. Lett. 14, 305 (1965)). Comparing them one can see that while for resistive transition T_c peaks to 0.4 K, for bulk magnetization the maximum T_c is 0.3 K). The difference between bulk and resistive T_c s was discussed in detail in recent literature (See for example Fig. 1 ref. 36 and Fig. 3 in arXiv:1804.07067). The authors should amend their paper by comparing their resistive T_c s with the resistive T_c of of Nb-doped and oxygen-reduced STO.

Response

We used the data in Ref. 20 (Ref. 23 in the previous manuscript) because it is the first (and one of the two) papers in the literature supplying the numeric values of both the critical temperature T_c and the carrier densities n for SrTiO_{3- δ} . However, as the Reviewer #1 pointed out, the T_c values in Ref. 20 were the inductively-determined values. Large discrepancies between the resistive and inductive T_c have been reported in the literature (*e.g.*, Refs. 20, 21, 28, and 34), and we wrote in page 6 of the previous manuscript as “Putting aside this issue, we used the resistive T_c in this study for fair comparison with the values of T_c in the literature.” Nevertheless, as the Reviewer #1 pointed out, we had made the “fallacious comparison” of the data in Ref. 20 with ours. This had been due to our simple misunderstanding, therefore **we have revised Fig. 3a to compare only the resistive T_c of SrTiO_{3- δ} from Ref. 21 as the Reviewer #1 suggested.**

But, for the comparison, **we need to change our definition of the resistive T_c** . Our previous T_c was defined as the temperature where the resistance becomes zero (see the previous Supplementary Information.) However, all the data in the literature, the values of T_c were defined as the temperature corresponding to the mid-point of the resistance drop for the superconducting transition. Therefore, we decided to **change all our T_c values throughout the manuscript to the newly defined values of mid-point T_c** for the fair comparison. (Accordingly, we have modified the description of the T_c definition in the revised Supplementary Information.) The differences of our old and new T_c values are small, thus **we are not necessary to change the discussion significantly in any parts of our manuscript.**

The Reviewer #1 also suggested that we should compare our T_c values to the recent publications as well, such as Ref. 31 and Ref. 35. We indeed agree with the suggestion because Ref. 21 was reported in 1965, which may be too old to be trustworthy for the present general readership. Thus, we have adopted the resistive T_c of SrTiO_{3- δ} and SrTi_{1- x} Nb _{x} O₃ from Ref. 31 and Ref. 35, and the comparison of those data are exhibited in the revised Fig. 3a.

If we compare our data with those obtained by modern and more accurate measurements in

Ref. 31 and Ref. 35, the superconducting dome of our $\text{Sr}_{1-x}\text{La}_x\text{TiO}_3$ samples shows much higher values of T_C than the values of the new $\text{SrTiO}_{3-\delta}$. However, both the superconducting dome of the old $\text{SrTiO}_{3-\delta}$ from Ref. 21 and that of new $\text{SrTi}_{1-x}\text{Nb}_x\text{O}_3$ from Ref. 35 seem to coincide with that of our $\text{Sr}_{1-x}\text{La}_x\text{TiO}_3$ in the smaller carrier-density region. Moreover, the former two show the optimal T_C in larger carrier-density region, where the values of T_C of our $\text{Sr}_{1-x}\text{La}_x\text{TiO}_3$ drop down rapidly. To be honest, we have no confident explanation beyond speculation why the old and new $\text{SrTiO}_{3-\delta}$ behave differently, and why the old $\text{SrTiO}_{3-\delta}$ behaves rather similar to new $\text{SrTi}_{1-x}\text{Nb}_x\text{O}_3$. These are quite interesting topics to be confirmed and clarified by further investigations.

We think the Reviewer #1 did not intend to request that we should clarify those differences in the revised manuscript, because the Reviewer #1 mentioned the important discovery in our work is (not there but) the large enhancement of T_C by the oxygen isotope exchange. Therefore, we simply show the comparison of our data with those in the literature, and leave the discrepancies as problems for the following detailed research.

Comment 2, Reviewer #1

2. La substitution and ferroelectricity- The following statement in the abstract appears to be inaccurate or unjustified: La substitution drives the system towards the ferroelectric QCP. Measurements of dielectric constant and Raman spectroscopy have shown that substituting 16O with 18O or Sr with Ca makes the solid ferroelectric. Is there any evidence that this is also the case by substituting Sr with La? Is there any evidence of the hardening of the soft TO mode in the La-substituted samples? Since the authors have performed Raman scattering, they can address this question.

Response

We fully agree with the concern raised by the Reviewer #1. We have no experimental evidences to directly corroborate “La substitution drives the system towards the ferroelectric QCP.” What we intended to advocate is that a theoretical model [Ref. 18] clearly predicts the existence of QCP in small carrier-density region of our $\text{Sr}_{1-x}\text{La}_x\text{TiO}_3$, even though the experimental data are all in the rather larger carrier-density region. The prediction corresponds to the appearance of $\Gamma < 1$ region shown in Fig. 3c and 3d.

However, we agree with the Reviewer #1 that it may be an overstatement to conclude as if this could be a firm evidence of the ferroelectric QCP. Therefore, **we have modified all the corresponding expression in the previous manuscript, and have mentioned in the revised manuscript that this QCP appearance is a prediction from a theoretical point of view.** We believe this modification will not discredit the worth of our manuscript.

On the other hand, although we will not stick to defending ourselves, the prediction of QCP is not simply a bizarre allegation. We have already discussed the appearance of QCP in the Supplementary Information, where we considered “the suppression of the ferroelectricity due to the large quantum fluctuation may be reduced by the Ca and Ba doping as an impurity. As an analogue to $\text{Sr}_{1-x}\text{Ba}_x\text{TiO}_3$ and $\text{Sr}_{1-x}\text{Ca}_x\text{TiO}_3$ systems, we can reasonably assume that the $\text{Sr}_{1-x}\text{La}_x\text{TiO}_3$ system should have a ferroelectric QCP.”

Unfortunately, our instrumental set-up for the Raman scattering experiment does not have a proper low-frequency-measurement option to conduct the requested experiment by the Reviewer #1. Moreover, our $\text{Sr}_{1-x}\text{La}_x\text{TiO}_3$ samples are quite metallic even for the smallest carrier-density sample. Hence, we wonder low-frequency Raman spectra might be smeared out by the large Drude contribution.

Therefore, we planned other experiments. We have already started collaborations with several other research groups to do precise x-ray and neutron scattering measurements, second

harmonic generation measurements, and so on, to investigate the long-range order as a direct evidence of the ferroelectricity even in the metallic $\text{Sr}_{1-x}\text{La}_x\text{TiO}_3$ samples. These projects are beyond the scope of this work, and will be reported in the near future.

Comment 3, Reviewer #1

3. Isotopic dependence of the magnitude of the T-square prefactor- The authors report that A become larger with ^{18}O substitution. However, what one sees in Fig.2b is an enhancement in resistivity well above the temperature window where resistivity plotted vs. T^2 is linear. The figure should be zoomed up to make this clear and the discussion should distinguish between the low-temperature T-square and the high-temperature T-cube (reported and discussed in Phys. Rev. 155, 796 (1967); NPJ Quantum Materials 2: 41 (2017) and arXiv:1806.05775 (2018)) regimes.

Response

We have been taken by the deep insight of the Reviewer #1. By following the advice of the Reviewer #1, **we have changed the range of the horizontal axis of Fig. 2b in the revised manuscript** to distinguish the low-temperature T^2 behaviour from the high-temperature T^3 behaviour. Thus, the enhancement of the A factor by the ^{18}O exchange becomes more apparent in the Resistivity- T^2 plot as shown in the new Fig.2b.

Comment 4, Reviewer #1

4. Mobility vs. carrier concentration in La-substituted strontium titanium- In thin films of La-doped Strontium titanate, carrier mobility can become as large as $30000 \text{ cm}^2/\text{V.s.}$ (See Nat. Mater. 9, 482 (2010)). The authors should comment on the fact that according to their Fig. 1D, the residual resistivity of their most underdoped sample ($x=0.0003$) appears to point to a much lower mobility. Does this mobility fit in to the general trend of mobility vs. carrier concentration seen in n-doped strontium titanate (See Fig. 4 in J. Phys.: Condens. Matter, 27 375501 (2015)).

Response

In the paper that the Reviewer #1 raised (Son *et al.* [Ref.24]), $\text{Sr}_{1-x}\text{La}_x\text{TiO}_3$ samples are thin films synthesised by molecular beam epitaxy (MBE) method. The paper advocates “In semiconductor physics, MBE is widely established as the deposition method that produces the highest mobility structures. It is a low-energetic, high-purity technique that allows for low defect densities and precise control over doping concentrations and location.” The MBE $\text{Sr}_{1-x}\text{La}_x\text{TiO}_3$ film is a very rare example in the long history of the research for the SrTiO_3 -related thin films. Moreover, the MBE film is two dimensional, while our $\text{Sr}_{1-x}\text{La}_x\text{TiO}_3$ sample is three dimensional. Thus **we think it is not appropriate to compare the mobility of our bulk $\text{Sr}_{1-x}\text{La}_x\text{TiO}_3$ with that of such a special thin film at the present stage of our research.**

On the other hand, another paper that the Reviewer #1 presented [Ref. 25] is quite intriguing. We have plotted the mobility of our $\text{Sr}_{1-x}\text{La}_x\text{TiO}_3$ on the Fig. 4c of Ref. 25 (See the figure below.) The purple solid circles are the mobility of our $\text{Sr}_{1-x}\text{La}_x\text{TiO}_3$. We have also added the $\text{Sr}_{1-x}\text{La}_x\text{TiO}_3$ data reported by Suzuki *et al.* [Ref.27] on the plot. Then, all the three $\text{Sr}_{1-x}\text{La}_x\text{TiO}_3$ data (Son *et al.*, Suzuki *et al.*, and ours) are located between “the general trend of mobility vs. carrier concentration” lines of $\text{SrTi}_{1-x}\text{Nb}_x\text{O}_3$ (orange) and $\text{SrTiO}_{3-\delta}$ (green). Mobility vs. carrier concentration relationship is an extremely interesting topic, but we really need much more delicate control of the sample quality, and is beyond the scope of this manuscript. **We**

have added this plot to the Supplementary Information and discussed the issue to some extent to encourage further detailed research in the near future.

We trust the Reviewer #1 would be satisfied with all our detailed replies above and will recommend publication of our work.

Reply to Reviewer #2

We are very happy to know that the Reviewer #2 could understand our work as we really expected; *i.e.*, “the authors are presenting a new materials pathway towards exploring the mysterious superconductivity in doped strontium titanate”, and our “studies will help the community distinguish the extrinsic and the intrinsic aspects of this rich problem”. On the other hand, we regret that our manuscript had contained some ambiguity and caused the Reviewer #2 a little misunderstanding. We would like to modify the ambiguous points to reflect the remarks made by the Reviewer #2.

Comment 1, Reviewer #2

1. In the abstract and then in the main text the authors write that lanthanum substitution both dopes STO and drives it towards a ferroelectric quantum critical point. They also note that La-doped STO crystals have not been previously studied systematically to date. Transport and Hall measurements support their statement about doping. It is known that oxygen substitution and application of external pressure can drive STO to its ferroelectric quantum critical point, but I do not see a reference or data to support the authors assertion here that La-substitution has a similar effect. More generally it would be very useful, particularly for comparison to previous work, to have a phase diagram for La-doped for oxygen-16 before considering the isotopically mixed case.

Response

We understand that the Reviewer #2 had a similar concern as that of the Reviewer #1, and again we agree with the Reviewer #2 completely. As described in the reply to the Comment 2 of the Reviewer #1, we are afraid that our claim in this manuscript was misunderstood. It is simply that we revealed “a theoretical model has predicted the existence of QCP of our $\text{Sr}_{1-x}\text{La}_x\text{TiO}_3$, even though our experimental data are limited only in the larger carrier-density region.” We think the claim itself is not necessary to be corroborated at this initial step by any experimental evidences. Motivated by this prediction, we are now working to find out the appearance of the long-range order as a direct evidence of the ferroelectricity even in the metallic $\text{Sr}_{1-x}\text{La}_x\text{TiO}_3$ samples by several kinds of special experiments. These are indeed the next steps of our research.

The phase diagram we showed in Fig. 3a in the previous manuscript was mainly for $\text{Sr}_{1-x}\text{La}_x\text{TiO}_3$ without ^{18}O exchange. $\text{Sr}_{1-x}\text{La}_x\text{TiO}_3$ with ^{18}O exchange has only five points (dark blue and light blue solid squares) in the phase diagram, thus, as we agree with the Reviewer #2, it is quite ambiguous to determine the phase boundaries for ^{18}O exchanged samples.

Therefore, **we have modified the Fig. 3a in the previous manuscript into Fig. 3a and Fig. 3b in the revised manuscript.** We have removed the fitting results from the Fig. 3a in the revised manuscript and added guide-to-the-eye lines for clear comparison of our $\text{Sr}_{1-x}\text{La}_x\text{TiO}_3$ without ^{18}O with $\text{SrTiO}_{3-\delta}$ and $\text{SrTi}_{1-x}\text{Nb}_x\text{O}_3$ in the literature. The fitting curves are moved to Fig. 3b in the revised manuscript. As mentioned above, the T_C values of our $\text{Sr}_{1-x}\text{La}_x\text{TiO}_3$ are sufficiently fitted by the theoretical curve which “predicts” a possible appearance of the ferroelectric QCP around $3 \times 10^{18} \text{ cm}^{-3}$. However, the old $\text{SrTiO}_{3-\delta}$ data [Ref.21] are not well fitted by a single theoretical curve for the larger carrier-density region. We have only three data points for $\text{Sr}_{1-x}\text{La}_x\text{Ti}({}^{16}\text{O}_{1-z}\text{}^{18}\text{O}_z)_3$, thus the fitting is not sufficient. But they clearly suggest the enhancement of T_C due to the QCP in larger carrier-density region.

Comment 2, Reviewer #2

2. The authors observe AT^2 behavior of the resistivity and ascribe it to Lifshitz transitions, as was done earlier by Lin and coworkers. However it’s not clear how this interpretation accounts for the observed temperature- dependence. It would be very helpful if the authors could expand their A vs. carrier density plot to compare their results with those found in other doped STO systems.

Response

We entirely agree with the Reviewer #2. We wrote in the page 4 of the previous manuscript:

“Therefore, the carrier density of our $\text{Sr}_{1-x}\text{La}_x\text{TiO}_3$ is too small for Umklapp scattering, indicating that the $\rho \sim AT^2$ behaviour up to high temperatures may be related to other mechanisms [18]. However, as we discuss later, the behaviour of the coefficient A appears to reflect the Lifshitz transition, which is the change of the number of Fermi surfaces that occurs by changing the carrier density. Therefore, we wrote the value of A is related to the fermiology of SrTiO_3 , which is intriguing.”

It is clear that the evolution of the effective mass changes at Lifshitz transition since the new Fermi surface is added. If $\rho \sim AT^2$ behaviour is originated in the normal electron-electron (Umklapp) scattering of the Fermi liquid theory, the relationship $A \propto m$ is valid. Therefore, by these two arguments, it is natural to observe a change of the A - n relation (derivative) at the Lifshitz point.

However, there is no reason for the opposite to be true. We simply wanted to mention that it is intriguing why we observed $\rho \sim AT^2$ behaviour without the Umklapp scattering. Furthermore, we wondered why this coefficient A seems to change at the Lifshitz point.

We regret that our description in the previous manuscript was not clear. Thus, **the description has been modified in the revised manuscript to emphasise this is a very interesting problem to be clarified in the future investigation.**

To meet the request of the Reviewer #2, **we have added data of other doped SrTiO₃ systems to the A vs. carrier density n plot of Fig. 2c in the revised manuscript.** Then, it becomes much clearer that there is actually a kink in the A vs. n plot around $n \sim 3.8 \times 10^{19} \text{ cm}^{-3}$; we would like to thank the Reviewer #2 for the insightful suggestion. It is very mysterious why this value of n corresponds to the critical carrier density of the Lifshitz transition, where the Fermi energy enters the third t_{2g} band [Ref. 15 and Ref. 30].

Comment 3, Reviewer #2

3. Naively one expects the superconducting transition temperature to be enhanced across a Lifshitz transition due to the availability of additional states. Mysteriously the opposite (T_C decreases) occurs in the measurements by Lin *et al.*, and it has been suggested that this behavior may be due to disorder. What is the behavior of T_C as a function of n here as bands cross the Fermi level?

Response

This is one of the most interesting and mysterious points of the doped SrTiO₃ systems. Lin *et al.* reported that T_C shows a peak at the first Lifshitz point (one band to two bands transition) at $n \sim 1 \times 10^{18} \text{ cm}^{-3}$, while T_C decrease toward the second Lifshitz point (two bands to three bands transition) at $n \sim 3 \times 10^{19} \text{ cm}^{-3}$ [Ref. 30]. As the Reviewer #2 commented, the Ref. 30 concluded that these are caused by the disorder, but it is not simply acceptable because the old SrTiO_{3- δ} samples in Ref. 21 showed completely opposite. In Ref. 21, T_C seemed to be diminished (not to show a peak) around the first Lifshitz point, and for increasing n towards the second Lifshitz point, T_C increases smoothly without a drop (see our revised Fig. 3a).

Unfortunately, our sample does not reach to the carrier-density range of the first Lifshitz point. However, at the second Lifshitz point, there is no anomaly in T_C as a function of carrier density (see our revised Fig. 3a). We understand the Reviewer #2 raised a very interesting question, but we cannot conclude anything on this mysterious issue. This point should be also elucidated by further detailed studies.

Comment 4, Reviewer #2

4. In Figure 3 the authors have compared their T_C vs n results to several previous studies. They suggest the importance of ferroelectric quantum criticality in maximizing the superconducting transition temperature. Their conclusions are similar to those from recent pressure-driven Nb-doped STO measurements (arXiv 1801.08121) and links with this work should be discussed.

Response

We were not aware of the manuscript in arXiv. Now, we have added the arXiv manuscript to our revised manuscript as Ref. 41. In Ref. 41, it is proposed that, if the carrier-density n is above a threshold n_{th} in the order of 10^{18} cm^{-3} , the value of T_C may increase when the system approaches QCP. If n is below n_{th} , T_C may decrease when the system approaches QCP. This is interesting because we have wondered whether T_C may be actually enhanced near QCP. The theoretical model [Ref. 18] that we adopted in this work has deduced QCP from our experimental

data, as well as an enhancement of T_C near QCP (see the dotted lines in Fig. 3b in the revised manuscript).

As the Reviewer #2 considered, this T_C enhancement near QCP in our $\text{Sr}_{1-x}\text{La}_x\text{TiO}_3$ work may be corroborated by Ref. 41 indirectly, because the QCP of our $\text{Sr}_{1-x}\text{La}_x\text{TiO}_3$ is predicted to be above $n_{\text{th}} \sim 10^{18} \text{ cm}^{-3}$. However, it should be also noticed that the work in Ref. 41 was done with a fixed carrier density while applying the pressure, but in our $\text{Sr}_{1-x}\text{La}_x\text{TiO}_3$ work, we approached QCP by changing the carrier density. We are afraid that this might have been misunderstood, so that **we have modified Fig. 3c in the previous manuscript (Fig. 4 in the revised manuscript)**.

We would like to thank all the valuable comments of the Reviewer #2. We hope our replies are satisfactory and the Reviewer #2 will recommend publication of our work.

Reply to Reviewer #3

We would like to acknowledge the Reviewer #3 for giving us several important questions. Although we are afraid that we may misunderstand the words of the Reviewer #3 that our manuscript “to be more sophisticated one before the review”, we tried to do our best to give a detailed reply addressing all the points raised by the Reviewer #3.

Comment 1, Reviewer #3

(1) First, the authors have not given the precision of the experiments. At least, they should give error bars in all the figures. Significant digits for all the data should be given in the text and figures, because very small change in the resistivity is discussed throughout the article. The reviewer cannot be convinced that all results are worthy for the detailed discussions.

Response

The resistivity ρ around 5 K of our $\text{Sr}_{1-x}\text{La}_x\text{TiO}_3$ is in the range of as small as $10^{-4} \Omega\text{cm}$. This value is obtained from the measured resistance R in the range of $\text{m}\Omega$ as $\rho = Rtw/l$, where the distance $l \sim 7 \text{ mm}$ between the voltage-probe electrodes along the current direction, the sample width $w \sim 0.3 \text{ mm}$ and the sample thickness $t \sim 0.5 \text{ mm}$ were measured by using a microscope with a scale of 40 digits per 1 mm, *i.e.*, the error is $1/40 \text{ mm}$. The value of R is in the range of $\text{m}\Omega$; therefore it is accurately measured by conventional instruments, *e.g.*, the resistance bridge or the digital voltmeter that in general has a resolution of $1\text{--}10 \mu\Omega$. Therefore, the approximate error of ρ is $\Delta\rho \simeq [1/100 + (1/0.5 + 1/0.3 + 1/7) \times (1/40)] \simeq 0.15$.

However, we did not discuss such a “very small change in the resistivity” throughout our manuscript. Thus, we think it is not necessary to add 15% error bars to our resistivity data.

In fact, we discussed values of only three physical quantities in the manuscript: superconducting critical temperature T_C , carrier density n , and coefficient A of the $\rho \propto AT^2$ behaviour. Therefore, we think the ambiguity, which the Reviewer #3 pointed out, may be for the values of T_C . As described in the reply to the Comment 1 of the Reviewer #1, we have changed the definition of T_C in the revised manuscript. Indeed, the definition of T_C is not common among researchers, and we believe it is quite necessary to define the method of determining the value of T_C . Considering such a different definition of the value of T_C , **we have attached “error bars” to the values of T_C in the revised manuscript; encompassing from the onset to the end value of the T_C drop**. We would like to emphasise that this is not due to any artefacts

of the measurements. However, the broad superconducting transition may reflect the quality of the sample, so we think it is a good idea to add the “error bars” to T_c , as the Reviewer #3 suggested.

As for the values of carrier density n , they were determined by the Hall effect measurement and all the values of the Hall voltage were in the range of mV or larger. Furthermore, the magnetic field dependence of the Hall voltage (*i.e.*, Hall resistance) was sufficiently linear and we had no problem of deducing the Hall coefficient (see the discussion in the Supplementary Information). Therefore, the error bar of the carrier density is negligibly small. Actually, as shown in the inset of the Fig. 1b in the main text, the carrier density deduced from the Hall effect is in very good accordance with the concentration of La. For our purpose of this work, this level of estimation of the carrier density is sufficient.

Finally, for the values of the coefficient A , they were determined by the least-square fitting of the experimental data. Because the temperature range, where the $\rho \propto AT^2$ relation is valid ($\gtrsim 80$ K), is more than several times wider than our fitting range (< 40 K), the fitting results are almost independent of the small change of the fitting range to estimate the error bar. In addition, we only use the value of A in the logarithmic plot, so that the $\sim 15\%$ error bar, corresponding to the error bar of ρ , is as small as the size of the markers in Fig. 2c.

Comment 2, Reviewer #3

(2) Sample preparation: In supplemental section. The authors have determined the isotope composition of the sample by the Raman spectroscopy. The authors have given no direction of the sample and optical setup used for the spectrum measurements. Also, reasonableness of the isotope composition determined by the Raman peak shift should be confirmed by comparing the result done by other method. Cross checking should be done to confirm the accuracy of the experiments.

Response

In our measurement, the Raman spectra were collected by the conventional backscattering geometry with unpolarised detection, therefore, it does not depend on the direction of the sample so much. However, to avoid an unnecessary confusion, **we have added a phrase “along the [100] direction” to the description of the Raman spectroscopy in the Supplementary Information.**

We asked the EAG laboratoriesTM (Sunnyvale, USA) to examine the $^{18}\text{O}/(^{16}\text{O}+^{18}\text{O})$ ratio of our $x \sim 0.0035, z = 0.57$ single crystal of $\text{Sr}_{1-x}\text{La}_x\text{Ti}(^{16}\text{O}_{1-z}^{18}\text{O}_z)_3$ by the dynamic secondary ion mass spectroscopy (SIMS). The value of z deduced by the SIMS measurement is $z = 0.65 \pm 0.10$. Since no discussion in our previous/revised manuscript requires the precise value of z , the SIMS result has guaranteed that the values of z deduced from our Raman spectroscopy are fairly trustworthy; thus, we keep using the latter values in the revised manuscript. **The result of SIMS is described in the revised Supplementary Information.**

Comment 3, Reviewer #3

(3) Fig.2(c): The oxygen-isotope-exchanged sample seems lying on a straight line. Does oxygen-isotope-exchanged system show a Lifshitz transition or not?

Response

This is a very interesting question and we really want to clarify it. As we replied to the Comment 3 of the Reviewer #1, the A values of ^{18}O -exchanged samples are larger than that of

non-exchanged samples even if we consider the possible 15% error bars. However, as already mentioned in the previous manuscript, we have only three points of the A values for ^{18}O -exchanged samples, and thus we cannot conclude whether the Lifshitz transition is apparently seen in the A vs. n plot for the ^{18}O -exchanged samples. We know this could be clarified by increase more data points of ^{18}O -exchanged samples on the plot, and the results would also give a clue to answer to the Comment 2 of the Reviewer #2.

Comment 4, Reviewer #3

(4) Fig.2(d): The result of AC susceptibility should be moved to Supplemental sections. It is confusing to show as an insert.

Response

This is an important plot as related to the Comment 1 of the Reviewer #1, however, as even the Reviewer #3 was confused, this may cause a similar confusion to the general readership. Thus, **we decided to remove the inset of showing the inductive- T_C and discuss this plot in the Supplementary Information.** Taking this opportunity, we measured the inductive- T_C again with more tight-rolling coils to the sample, and the results of the inductive- T_C was a bit increased, which, however, does not affect to the discussion in the main text.

Comment 5, Reviewer #3

(5) Fig.3(c): What is the definition of ferroelectricity? The authors should explain the detail of the meaning although this figure is too schematic. What is the physical meaning of the straight lines? How did the authors draw the OCL curves? These points should be explained clearly.

Response

Many papers in the field of superconductivity in SrTiO_3 recently discuss on the possibility of the coexistence of ferroelectricity and superconductivity of this material (for example, Ref. 18, Ref. 28, and Ref. 47). The quantum critical line (QCL) is the schematic representation of the coexistence, *i.e.*, the boundary of ferroelectric & metallic state and the normal metallic state. In general, ferroelectricity should rapidly disappear in the metallic state. However, stronger ferroelectricity may remain more in the metallic state; thus the boundary (QCL) is schematically written like that (see Ref. 18).

It is considered in general that the “ferroelectricity” in the metallic state means a long-range order of the polar lattice deformation as is easily seen when the system is a conventional ferroelectric insulator. But the polarisation of the ferroelectric metal is not observed because of the screening by the metallic carriers.

The straight lines represent chemical substitution (including the ^{18}O -exchange and oxygen-deficiency creation) processes. Of course, they are not necessary to be the straight lines, but in the schematic diagram, there is no merit to give hypothetical curvatures to those lines and make the simple view more complicated.

We realised that our old Fig. 3c in the previous manuscript may not be easily understood. Therefore, **we have modified the figure and is now Fig. 4 in the revised manuscript.** We have totally modified the description of the Fig. 4 in the main text.

We hope we have cleared up all the misunderstandings of the Reviewer #3 and our work will be recommended publication.

Summary of changes in the manuscript file.

According to the point by point replies to the questions and comments raised by the Reviewers described above, we have revised our manuscript carefully. **All the changes in the Main Manuscript file are highlighted by red letters.** Therefore, here we give some additional or further explanations for the changes.

1. A description on the sample direction has been added in the Raman spectroscopy part in the Supplementary Information.
2. SIMS results have been described in the Supplementary Information.
3. We have added the mobility comparison in the Supplementary Information.
4. The range of Fig. 2**b** has been changed.
5. More data in the literature have been added to Fig. 2**c**.
6. The inset of Fig. 2**d** in the previous manuscript has been removed and discussed in the Supplementary Information with updated data.
7. As described in the response to the Reviewer #1, we have re-defined our T_C from the previous zero-resistance definition to the new mid-point definition. Therefore, all the values of T_C of our work have been modified to new values throughout this manuscript.
8. The T_C definition part in the Supplementary Information has been revised accordingly.
9. The Fig. 3**a** in the previous manuscript has been divided into Fig. 3**a** and Fig. 3**b** in the revised manuscript.
10. The “error-bar” in Fig. 3**a** represents the range of the T_C drop from the onset to the end.
11. Fig. 3**b** in the previous manuscript has also been modified as Fig. 3**c** in the revised manuscript, because we did the fitting again for the new T_C data.
12. The Fig. 3**c** in the previous manuscript has been changed and is now Fig. 4 in the revised manuscript.
13. We have added new references (Ref. 24, 25, 29, 31, 41, 42, and 44) in the revised manuscript and removed some in the previous manuscript. Therefore, the number of each reference has been changed.
14. After submitting the previous manuscript, we had some chances to have discussions with several researchers, especially, on the relationship between superconductivity and ferroelectricity. Although their opinions and comments cannot be reflected in this revised manuscript directly, we would like to thank all of them for their stimulating discussions. Thus, we have added their names in the acknowledgement.
15. We have corrected some typos and grammatical errors of the English of the manuscript. The corrections as well are highlighted in the revised manuscript with other major modifications mentioned above.

We would like to acknowledge again to all the reviewers for the essential questions. We hope they are satisfied with our detailed reply and will recommend publication of our work.

REVIEWERS' COMMENTS:

Reviewer #1 (Remarks to the Author):

The authors have done a very satisfactory job in addressing the reviewers' concerns. I think the paper is now ripe for publication.

Reviewer #2 (Remarks to the Author):

I am satisfied with the detailed responses and changes the authors have made to all the reviewers' comments/questions/suggestions. In my opinion this paper is now suitable for publication in Nature Communications.